# Phenotypic Detection of Hemin-Inducible Trimethoprim-Sulfamethoxazole Heteroresistance in *Staphylococcus aureus*

Dennis Nurjadi,[a] Quan Chanthalangsy,[a] Elfi Zizmann,[a] Vanessa Stuermer,[a] Maximilian Moll,[a] Sabrina Klein,[a] Sébastien Boutin,[a] Klaus Heeg,[a] Philipp Zanger[a,b]

[a]Department of Infectious Diseases, Medical Microbiology, and Hygiene, Heidelberg University Hospital, Heidelberg, Germany
[b]Heidelberg Institute of Global Health, Heidelberg University Hospital, Heidelberg, Germany

**ABSTRACT** Trimethoprim-sulfamethoxazole (SXT) is a valuable second-line antimicrobial agent to treat methicillin-resistant *Staphylococcus aureus* infections. Discrepancies between various antibiotic susceptibility testing (AST) methods for SXT susceptibility in *S. aureus* have been described. Here, we describe a hemin-inducible heteroresistance phenotype in *S. aureus*. We compared the results of the Vitek 2 AST on a set of 95 *S. aureus* clinical isolates with broth microdilution, disk diffusion using standard Mueller-Hinton agar, and disk diffusion using Mueller-Hinton agar supplemented with 5% horse blood (MHF). To investigate the potential clinical relevance of SXT heteroresistance, an *in vivo Galleria mellonella* infection assay was performed. All Vitek 2 SXT-susceptible (*n* = 17) isolates were concordant with AST results by other methods applied in this study. In 32/78 (41%) of Vitek 2 SXT-resistant isolates, we observed a heteroresistant growth phenotype on MHF. The heteroresistance phenotype was associated with the presence of *dfr* genes, encoding trimethoprim resistance. The addition of a hemin-impregnated disk in a double disk diffusion method on standard Mueller-Hinton agar was able to induce growth in the SXT zone of inhibition. An *in vivo* infection assay with *G. mellonella* suggested that the SXT heteroresistance phenotype resulted in lethality similar to that of the SXT-resistant phenotype. In this study, we describe a novel hemin-inducible heteroresistance phenotype in *S. aureus*. This heteroresistance phenotype may be missed by standard AST methods but can be detected by performing disk diffusion using Mueller-Hinton agar supplemented with 5% horse blood, commonly used for AST of fastidious organisms. This phenomenon may partly explain the discrepancies of AST methods in determining SXT resistance in *S. aureus*.

**IMPORTANCE** *Staphylococcus aureus* is one of most important pathogens in clinical medicine. Besides its virulence, the acquisition or emergence of resistance toward antibiotic agents, in particular to beta-lactam antibiotics (methicillin-resistant *S. aureus* [MRSA]), poses a major therapeutic challenge. Trimethoprim-sulfamethoxazole (SXT) is one of the effective antimicrobial agents of last resort to treat MRSA infections. Here, we report the detection of a SXT-heteroresistant phenotype which is inducible by hemin and can be detected using Mueller-Hinton agar supplemented with horse blood. Heteroresistance describes the presence or emergence of resistant subpopulations, which may potentially lead to inaccurate antibiotic susceptibility testing results and influence the success of antibiotic therapy.

**KEYWORDS** MRSA, *Staphylococcus aureus*, antibiotic susceptibility testing, co-trimoxazole, hemin, heteroresistance, trimethoprim-sulfamethoxazole

Antimicrobial resistance is an ongoing global concern which has led infectious diseases specialists to resort to older, less frequently used antibiotics. The antifolate trimethoprim-sulfamethoxazole (SXT) is orally available and an important and valuable antibiotic agent that is considered one of the antibiotics of last resort to treat infections due to methicillin-resistant *Staphylococcus aureus* (MRSA) (1–3). Over the past few years, SXT has

Address correspondence to Dennis Nurjadi, dennis.nurjadi@uni-heidelberg.de.

experienced a renaissance, as reflected in numerous efficacy studies involving SXT to treat *S. aureus* infections (4, 5). A recent randomized control trial demonstrated the noninferiority of SXT to vancomycin for the treatment of severe infections caused by MRSA, highlighting the importance of this agent for the treatment of MRSA (5).

SXT is a combination of two antifolate drugs, trimethoprim (TMP) and sulfamethoxazole (SMZ). Its antibacterial activity relies on the inhibition of two sequential key enzymes in the bacterial folic acid synthesis pathway (dihydrofolate reductase [DHFR] and dihydropteroate synthase [DHPS]), which can be bypassed in the presence of exogenous thymidine (6).

Since SXT resistance in *S. aureus* is on the rise (7–10), the accuracy of antibiotic susceptibility testing (AST) is essential for the success of antimicrobial therapy. Recently, various reports on the discrepancy of different AST methods for SXT have been reported independently (11–13), casting doubts on the accuracy and reliability of AST for SXT in *S. aureus*. Until now, there are no definite explanations for this discrepancy. In this study, we initially aimed to compare the performance of Vitek 2 with various solid testing media in determining phenotypic SXT susceptibility in clinical *S. aureus* isolates. In doing so, we encountered a heteroresistance phenotype which was inducible by hemin and can be detected by performing disk diffusion using Mueller-Hinton agar supplemented with 5% horse blood. Here, we present our findings of this heteroresistance phenomenon. Furthermore, we investigated the potential clinical relevance of these findings using an *in vivo Galleria mellonella* infection assay.

## RESULTS

We compared the performance of disk diffusion in determining SXT resistance in *S. aureus* using two commercially available Mueller-Hinton agar ($MH_{BD}$ and $MH_{BM}$) from two manufacturers (BD Diagnostics and bioMérieux, respectively) along with Mueller-Hinton agar supplemented with 5% horse blood (MHF). AST was performed additionally on MHF, since horse blood naturally contains the enzyme thymidine phosphorylase, which degrades thymidine in the culture medium, thus creating a low-thymidine environment (14). The presence of exogenous thymidine in the culture medium can overcome both TMP and SMZ activity, which may deliver false-resistant results (14, 15). Overall, 95 isolates were tested, 17/95 (18%) were SXT susceptible, and 78/95 (82%) were SXT resistant according to the Vitek 2 AST. The differences in the diameters of the zone of inhibition are displayed in Fig. 1a. All Vitek 2 SXT-susceptible isolates exhibited SXT susceptibility on all tested agar media ($MH_{BD}$, $MH_{BM}$, and MHF). Out of 78 isolates with Vitek 2 SXT resistance, only 3/78 (4%) tested resistant, while 9/78 (12%) tested susceptible (with increased exposure) on $MH_{BD}$. Similarly, only 2/78 (3%) tested resistant on $MH_{BM}$. On MHF, 5/78 (6%) tested susceptible (with increased exposure) and 2/78 (3%) tested resistant (Table 1).

**Phenotypic heteroresistance to trimethoprim-sulfamethoxazole.** In 32.6% (32/95) of the study isolates, we noticed visible colony growth within the SXT zone of inhibition on MHF, resembling heteroresistance growth characteristics, which was not present when using standard Mueller-Hinton agar as the testing medium (Fig. 1b). This phenomenon was observed in both disk diffusion and Etest on MHF. While most isolates with the heteroresistant phenotype (30/32; 93.8%) exhibited only slight growth (low density), two of the isolates (2/32; 6.3%) exhibited substantial growth (high density) within the inhibition zone (Fig. 1c). We did not observe the heteroresistance phenotype in any of the 17 Vitek 2 SXT-susceptible isolates, whereas 32/78 (41.0%) Vitek 2 SXT-resistant isolates exhibited the heteroresistance growth phenotype (Table 1). Of those isolates with discordant Vitek, and disk diffusion results (i.e., SXT resistant by Vitek 2 and SXT susceptible by disk diffusion), the heteroresistance phenotype was overrepresented (>50%) in isolates with an MIC of <2 mg/liter (Fig. 2). Of the 78 Vitek 2 SXT-resistant isolates, 44 (56.4%) isolates were nonheteroresistant and 2 (2.6%) exhibited the full (high-level) resistance phenotype.

**Molecular characteristics and stability of the heteroresistance phenotype.** There was no apparent association between the genetic background (multilocus sequence type [MLST]) of the isolates and the heteroresistance phenotype (see Table S2 in the supplemental material). All isolates exhibiting the heteroresistance phenotype were trimethoprim resistant and harbored an extrachromosomal *dfr* gene, either the trimethoprim resistance gene *dfrG* (23/32;

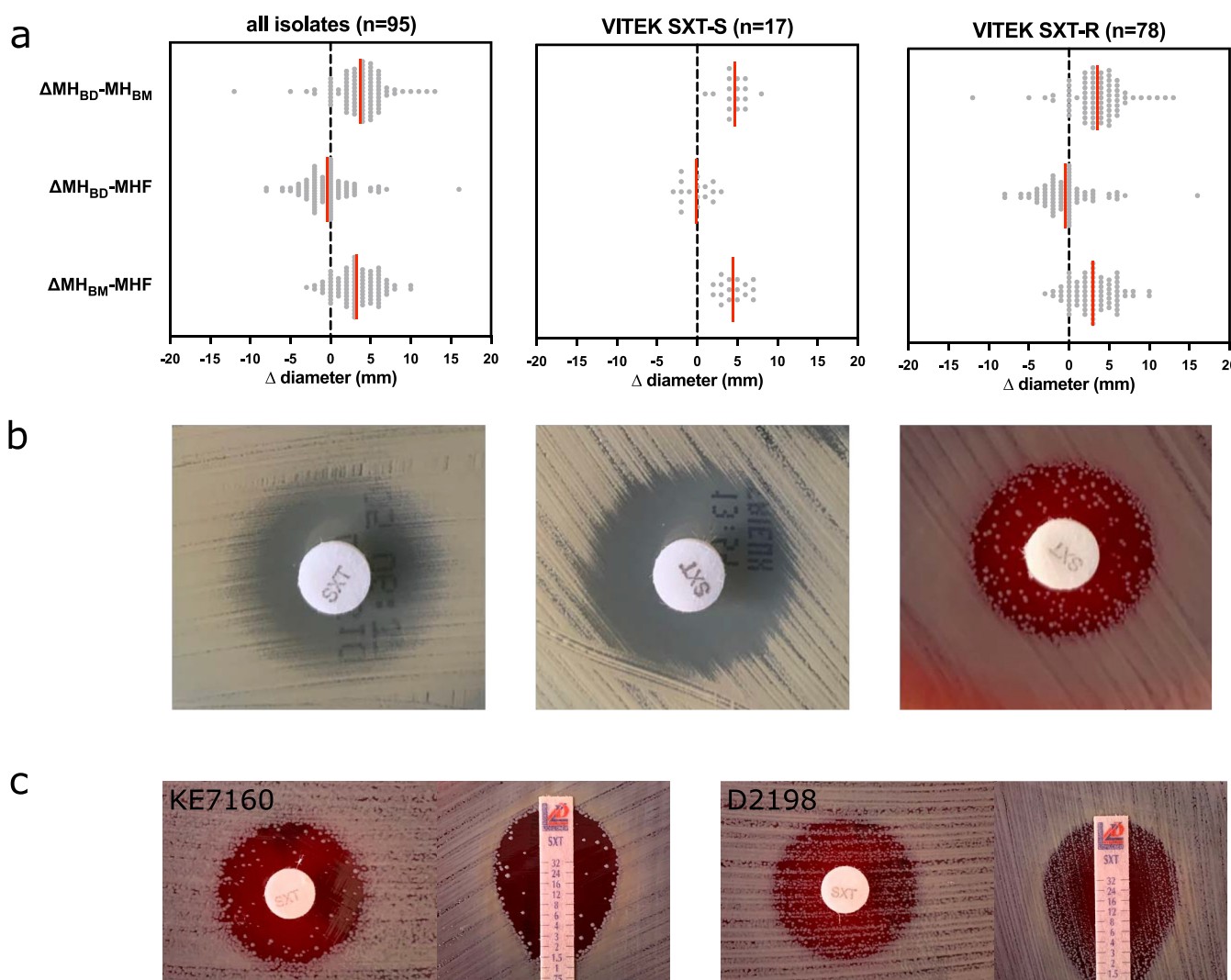

**FIG 1** Heteroresistance phenotype toward trimethoprim-sulfamethoxazole in *Staphylococcus aureus*. (a) Discrepancies between Vitek 2 and disk diffusion in susceptibility testing for trimethoprim-sulfamethoxazole (SXT). Disk diffusion was performed using three different media, Mueller-Hinton agar from Becton, Dickinson (MH_BD), Mueller-Hinton agar from bioMérieux (MH_BM), and Mueller-Hinton agar with 5% horse blood (MHF). SXT-S, SXT susceptible; SXT-R, SXT resistant. (b) Disk diffusion of phenotypically SXT-heteroresistant *S. aureus* KE7160 on various Mueller-Hinton agars. (c) Low-density heteroresistance phenotype (left panel) and high-density heteroresistance phenotype (right panel) in Kirby-Bauer disk diffusion and MIC test strip.

72%) or *dfrA* (9/32; 28%). The presence of *dfr* genes, however, is not an ideal determinant of the heteroresistance phenotype. Of the 44 nonheteroresistant isolates, 42 (95.5%) harbored extrachromosomal *dfr* genes (9/42 [21.4%] with *dfrA*, 29/42 [69.1%] with *dfrG*, 1/42 [2.4%] with *dfrK*, 1/42 [2.4%] with *dfrD*, and 2/42 [4.8%] with *dfrA* and *dfrG*).

Next, to test the stability of this phenotype, a daily subculture and subsequent repeated disk diffusion over 4 days (over 40 generations) without antibiotic pressure of a single colony from inside the SXT inhibition zone displayed a heteroresistance phenotype similar to that of the initial disk diffusion, suggesting that the heteroresistant phenotype is in fact stable and monoclonal (Fig. S1). To investigate if the heteroresistant phenotype is an artifact produced by the presence of thymidine in the testing medium, we performed a modified population analysis profile (mPAP) using a photometric approach (optical density at 590 nm [$OD_{590}$]) in the presence and absence of 5 IU/ml thymidine phosphorylase (see Fig. S3 in the supplemental material). The growth characteristics in the mPAP were consistent with the PAP of phenotypic heteroresistance as proposed by Andersson et al. (16). The addition of 5 IU/ml thymidine phosphorylase to the liquid medium did not have any effect on the PAP, suggesting that this phenomenon was independent of the thymidine content in the testing medium.

**TABLE 1** Concordance of trimethoprim-sulfamethoxazole Vitek 2 susceptibility testing with disk diffusion and broth microdilution in *S. aureus*[a]

| AST result[b] | Vitek 2 result | | | |
|---|---|---|---|---|
| | S (n = 17) | | R (n = 78) | |
| | n | % | n | % |
| Mueller-Hinton agar 1 (MH$_{BD}$)[c] | | | | |
| Susceptible (standard dosing regime) | 17 | 100 | 66 | 84.6 |
| Susceptible (with increased exposure) | 0 | 0 | 9 | 11.5 |
| Resistant | 0 | 0 | 3 | 3.9 |
| Mueller-Hinton agar 2 (MH$_{BM}$)[d] | | | | |
| Susceptible (standard dosing regime) | 17 | 100 | 76 | 97.4 |
| Susceptible (with increased exposure) | 0 | 0 | 0 | 0 |
| Resistant | 0 | 0 | 2 | 2.6 |
| Mueller-Hinton agar with 5% horse blood (MHF)[c] | | | | |
| Susceptible (standard dosing regime) | 17 | 100 | 71 | 91.0 |
| Susceptible (with increased exposure) | 0 | 0 | 5 | 6.4 |
| Resistant | 0 | 0 | 2 | 2.6 |
| Broth microdilution | | | | |
| Susceptible (standard dosing regime) | 17 | 100 | 44 | 56.4 |
| Susceptible (with increased exposure) | 0 | 0 | 32 | 52.6 |
| Resistant | 0 | 0 | 2 | 2.6 |
| Trimethoprim susceptibility (DD)[e] | | | | |
| Susceptible | 13 | 76.5 | 1 | 1.3 |
| Resistant | 4 | 23.5 | 77 | 98.7 |

[a]S, susceptible; R, resistant; AST, antibiotic susceptibility testing; DD, disk diffusion.
[b]AST was interpreted according to EUCAST clinical breakpoints (v11.0) using the following categories: susceptible (standard dosing regime), susceptible (increased exposure), and resistant.
[c]MH$_{BD}$ purchased from Becton, Dickinson, Germany.
[d]MH$_{BM}$ purchased from bioMérieux, Germany.
[e]Trimethoprim susceptibility was determined by disk diffusion on MH$_{BD}$, MH$_{BM}$, and MHF (all results were concordant). Phenotypic trimethoprim resistance is associated with Vitek 2 SXT resistance ($P < 0.001$ using chi-square test).

**The heteroresistance phenotype is inducible by hemin.** Reduced susceptibility to SXT has been described for the *S. aureus* small colony variant (SCV) (17), often involving either thymidine, menadione, or hemin auxotrophy. Both KE7160 and D2198 showed normal growth on MH agar and Columbia blood agar. Phenotypic testing did not indicate thymidine, menadione, or hemin auxotrophy (Fig. S4). Since the heteroresistance phenotype was present only on the MHF medium, not on standard MH agar, we compared the components of both culture media (from the same manufacturer) and identified defibrinated horse blood and 20 mg/liter β-NAD as additional supplements in the MHF. Since

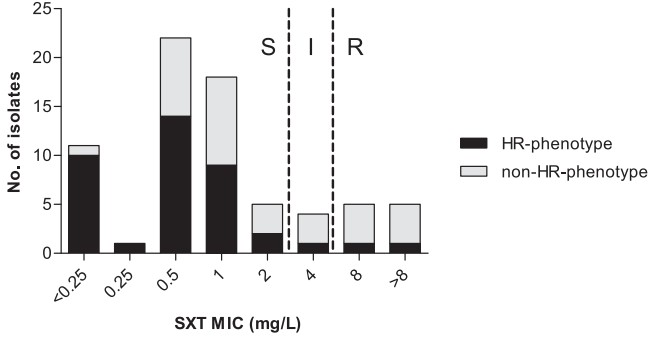

**FIG 2** Association between MIC and the trimethoprim-sulfamethoxazole heteroresistance (HR) phenotype for resistant isolates according to Vitek. The heteroresistance phenotype is overrepresented in isolates with an MIC of <4 mg/liter compared to isolates with an MIC of ≥4 mg/liter.

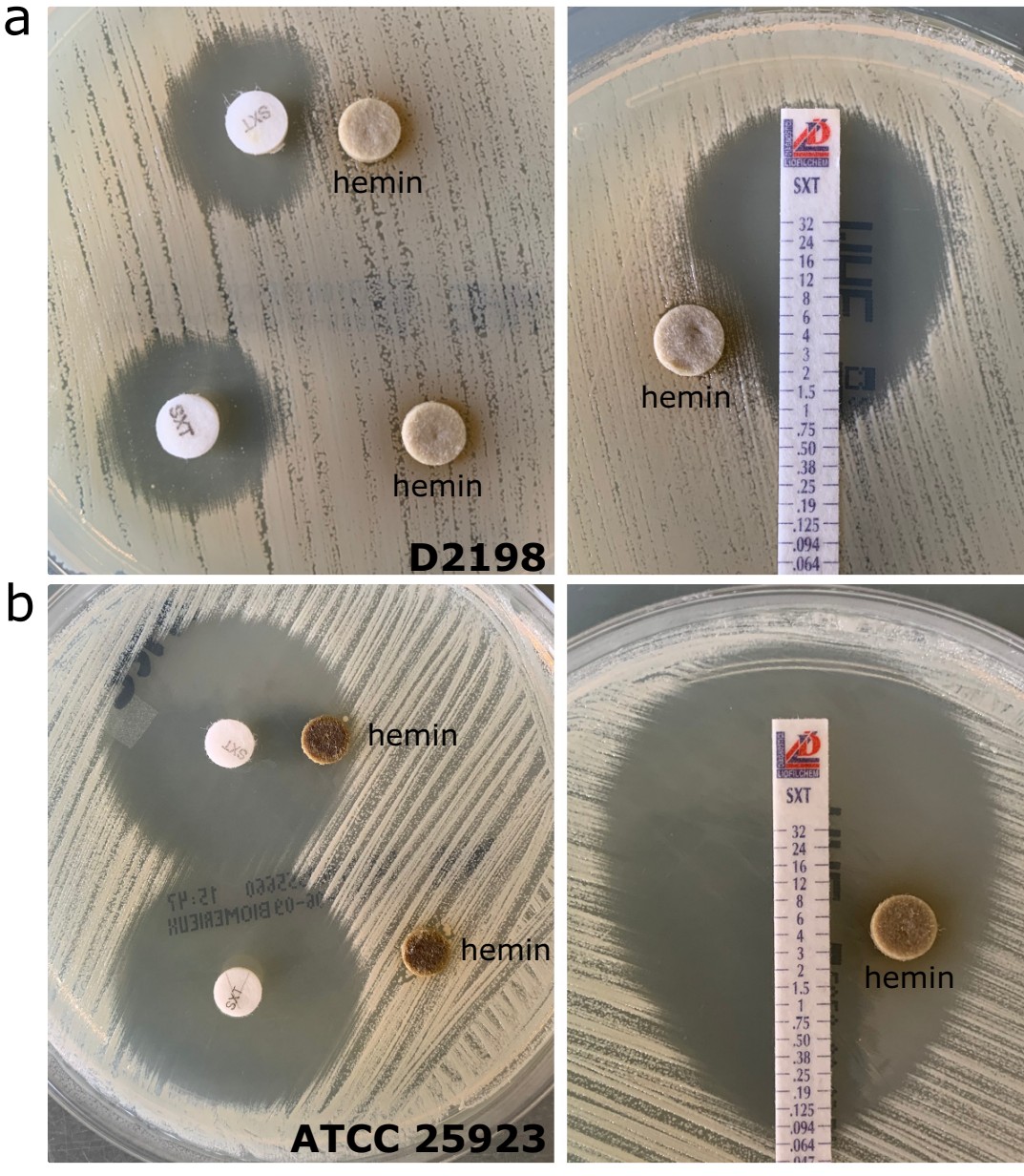

**FIG 3** Heteroresistance phenotype toward *Staphylococcus aureus* can be induced by hemin. (a) In a double disk diffusion using trimethoprim-sulfamethoxazole- and hemin-impregnated disks, there was a distortion of the inhibition zone due to growth within the inhibition zone of the SXT-heteroresistant isolate D2198 on standard Mueller-Hinton agar (left panel), which was also visible in the Etest (right panel). (b) The growth characteristics of SXT-susceptible ATCC 25923 was not affected by hemin. The variations in the color of the hemin disk may be the result of uneven distribution/diffusion into the agar medium or hemin saturation in the area surrounding the disks.

MHF is used for AST of fastidious organisms, such as *Haemophilus influenzae*, which required $\beta$-NAD (synonym, factor V) and hemin (synonym, factor X) to grow (18), we then tested if either substance induces phenotypic heteroresistance toward SXT using the double disk diffusion method. $\beta$-NAD did not affect the growth of either KE7160 or D2198 in the presence of SXT (data not shown). However, the presence of hemin distorted the SXT zone of inhibition in a manner similar to the inducible clindamycin resistance in *S. aureus* (data for hemin in D2198 are shown in Fig. 3; data for KE7160 are not shown) (19).

**Infection assay with *Galleria mellonella*.** To investigate the potential clinical implication of SXT heteroresistance for *S. aureus* infections, we performed an *in vivo* infection assay with *Galleria mellonella* to determine the outcome of infections with a heteroresistant strain

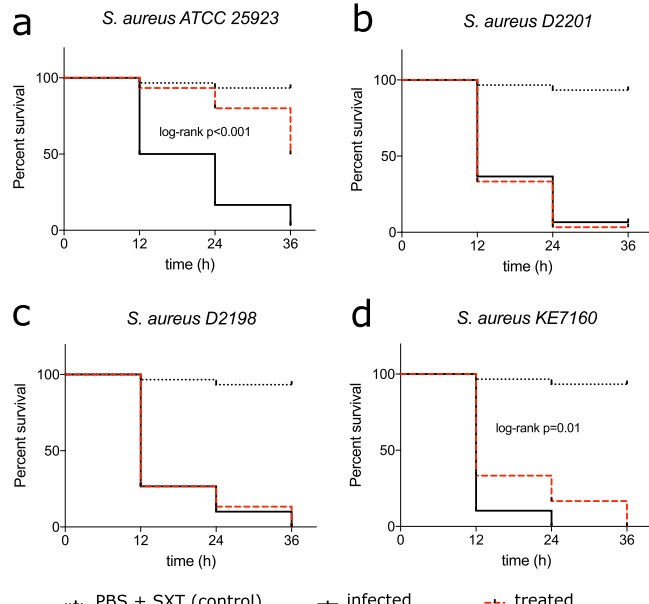

**FIG 4** Survival analysis after single-dose trimethoprim-sulfamethoxazole following *Staphylococcus aureus* infection in *Galleria mellonella*. Pooled data ($n$ = 30) from three independent experiments ($n$ = 10 each) are shown. For the infection, $5 \times 10^6$ CFU viable *S. aureus* was injected into the prodromal legs of the *G. mellonella*. In the treated group, a single dose of 10 mg/kg body mass of trimethoprim-sulfamethoxazole (SXT) was administered (PBS was administered in the nontreated group). (a to d) Infection with SXT-susceptible *S. aureus* ATCC 25923 (a), SXT-resistant *S. aureus* D2201 (b), and SXT-heteroresistant *S. aureus* D2198 (c), and KE7160 (d). Statistical significance was calculated using the log rank test (between infected and treated groups), and only significant values are displayed.

of *S. aureus* following SXT administration. D2201, an *S. aureus* strain with a high level of SXT resistance, was used as a positive control, and ATCC 25923 was used as a susceptible control. In initial experiments, we determined $5 \times 10^6$ CFU as the optimal infection dose to induce mortality in *G. mellonella* within 36 h postinfection (p.i.). The injection of a single dose of 10 mg/kg SXT (1:19) 2 h p.i. significantly increased survival/delayed mortality in ATCC 25923-infected *G. mellonella* larvae (log rank $P < 0.0001$) but not in D2201-infected larvae (Fig. 4). Infection with both low-density-growth (KE7160) and high-density-growth (D2198) heteroresistant *S. aureus* strains showed similar outcomes as infection with D2201, with most larvae dead by 36 h p.i. There was a slight delay in time to endpoint for KE7160 with statistical significance ($P = 0.01$); however, the outcome in terms of survival was comparable to that of D2201- and D2198-infected larvae (Fig. 4).

## DISCUSSION

Heteroresistance is a challenge for both clinical microbiologists and infectious diseases practitioners alike. Falsely interpreted susceptible microorganisms can jeopardize therapy success, leading to higher morbidity and mortality, while false-resistant organisms can lead to overuse of second-line and last-resort antibiotics. Hence, the accuracy and reliability of AST are essential to guide antimicrobial therapy management. Recently, discordance in SXT AST for *S. aureus* between the Vitek 2 system, disk diffusion, and broth microdilution have been reported (11–13). In our study isolates, we saw discrepancies between the Vitek 2 and disk diffusion AST for Vitek 2 SXT-resistant isolates, which is in line with the findings of Coombs et al, who found that Vitek-based diagnosis of resistance can often not be confirmed by other methods (11). In our case, over 40% of Vitek 2 SXT-resistant isolates were either SXT susceptible at the standard dosage or with increased exposure, with substantial colony growth within the zone of inhibition. In other studies reporting higher proportions of SXT resistance by Vitek 2, the authors reported that they could not detect any acquired genes or mutations, which may provide an explanation for this discrepancy. By testing TMP and SMZ separately, we found that the TMP-resistant phenotype and the

presence of extrachromosomal *dfr* is overrepresented in isolates of the Vitek 2 SXT-resistant group.

In this study, we report a hemin-inducible SXT heteroresistance phenomenon in *S. aureus*, which can be detected by performing disk diffusion on Mueller-Hinton agar with 5% horse blood. The concept of SXT heteroresistance in *S. aureus* is not completely novel (13, 20). In contrast to the findings by Coelho et al. (20) and Scholtzek et al. (13), our heteroresistance phenotype was visible only in the presence of hemin and SXT, which would remain undetected in standard AST by disk diffusion using conventional MH agar. In the studies by Coelho et al. and Scholtzek et al., the AST was not performed in a thymidine-free environment, so that slight growth or a hazy inhibition zone due to the presence of exogenous thymidine cannot be fully ruled out. Furthermore, Scholtzek et al. described in their study that the heteroresistant growth was visible after 48 h of incubation, which is definitively longer than the recommended incubation time for AST (18 h), so that SXT degradation is possible. In our study, the phenotypic detection of hemin-inducible SXT heteroresistance was performed using MHF. Horse blood naturally contains thymidine phosphorylase, so we are confident that the inducible resistant phenotype in our study was not an artifact due to exogenous thymidine in the testing medium. This was also confirmed by the results of the mPAP with and without thymidine phosphorylase.

A potential explanation for this heteroresistance phenomenon is the induction of the small colony variant (SCV) of *S. aureus*. This SCV phenotype is usually an expression of thymidine, menadione, or hemin auxotrophy, has been linked to reduced susceptibility to SXT (21), and is commonly encountered in *S. aureus* isolates from cystic fibrosis patients (22). Although the normal (wild-type) colony morphology on both Columbia blood agar and standard Mueller-Hinton agar, together with the negative auxotrophy results, does not support the SCV theory, we cannot definitely rule out the possibility that our heteroresistance phenotype was not the result of SCV emergence due to spontaneous morphological or transcriptional changes through stress conditions. Indeed, SCV and other phenotypic changes can occur spontaneously in *S. aureus* during replication (23), so that the observed heteroresistance can be the result of a spontaneous phenotypic switch or the emergence of a hemin-auxotrophic subpopulation. Although the exact underlying mechanism is still unclear, our data indicate that the heteroresistance phenotype was not lineage (MLST) specific and was associated with the presence of the TMP resistance determinant, extrachromosomal *dfr* genes. That being said, neither of these parameters (TMP resistance or extrachromosomal *dfr* genes) was a reliable parameter to predict or detect our heteroresistance phenotype, since a significant proportion of nonheteroresistant isolates also harbor extrachromosomal *dfr* genes and exhibit TMP resistance.

The idea of heterogeneous phenotypic resistance is not completely novel and has been described for several species previously (16, 24). In *S. aureus*, phenotypic heteroresistance has been described for SXT, $\beta$-lactams, and vancomycin (20, 25, 26). Even though the clinical implication of heteroresistance is still unclear, some studies have reported clinical significance of heteroresistance in *S. aureus* (27, 28). Our simplified *G. mellonella in vivo* infection model suggested that the outcome (survival) of infection with SXT-heteroresistant *S. aureus* was similar to that with SXT-resistant *S. aureus*, suggesting that SXT heteroresistance in *S. aureus* may have some clinical relevance. Further clinical investigations are needed to reliably determine the clinical significance of SXT heteroresistance in *S. aureus*.

Our study has limitations. We randomly chose 95 clinical isolates, and therefore the data presented here cannot estimate the real prevalence of heteroresistance in the general *S. aureus* population. However, over one-third of the study isolates exhibited SXT heteroresistance, and published data suggest that the occurrence of SXT heteroresistance is not a rare event (12). Furthermore, our proposed testing medium using MHF is not the generally recommended medium for *S. aureus* AST. Nevertheless, MHF is commercially available and is a guideline-conforming medium for AST of fastidious organisms. Despite the lack of consensus on the definition of heteroresistance, we adhered, whenever possible, to the recommended criteria (clonality, level of resistance, and stability) and nomenclature proposed by Andersson et al. in defining heteroresistance (29).

In conclusion, we demonstrated that hemin could induce monoclonal SXT heteroresistance in *S. aureus*, which can be detected using Mueller-Hinton agar supplemented with 5% horse blood. This phenomenon may partly explain the discrepancies in SXT susceptibility between various AST methods. In this study, we demonstrate the potential impact of heteroresistance in therapeutic failure. Therefore, this knowledge may be valuable for microbiological diagnostics, especially in the case of persistent *S. aureus* infections despite SXT therapy. However, further investigations are needed to determine the underlying mechanisms and evaluate the clinical significance of SXT heteroresistance in *S. aureus*.

## MATERIALS AND METHODS

**Microbiological methods.** Ninety-five characterized *S. aureus* clinical isolates from a prior study were chosen at random (30). Antibiotic susceptibility for all isolates was determined using Vitek 2 and disk diffusion using various media. AST for SXT by the Vitek 2 system (bioMérieux GmbH, Germany) was performed according to the manufacturer's protocol using the P654 AST card for staphylococci. Disk diffusion for trimethoprim (5 $\mu$g), trimethoprim-sulfamethoxazole (1.25 $\mu$g plus 23.75 $\mu$g) (Sensi-Disc, BD Diagnostics, Germany), and sulfamethoxazole (300 $\mu$g) (Oxoid, Germany) was performed according to EUCAST standards using two different Mueller-Hinton (MH) agar plates from BD Diagnostics (Germany) (MH$_{BD}$) and bioMérieux (Germany) (MH$_{BM}$) and Mueller-Hinton agar supplemented with 5% horse blood and 20% NAD (MHF) (bioMérieux, Germany). The zone of inhibition (in mm) was measured after 18 $\pm$ 2 h of incubation at 35 $\pm$ 1°C. *S. aureus* ATCC 25923 served as a control. An MIC test strip for SXT (Liofilchem, Italy) was used according to the manufacturer's instructions. MIC by broth microdilution was performed using a commercially available testing panel (Merlin Diagnostika GmbH, Germany). Antibiotic susceptibility was interpreted according to the EUCAST clinical breakpoints (v11.0). Double disk diffusion was performed using $\beta$-NAD-impregnated disks (Sigma-Aldrich, Germany) and 200 $\mu$g/ml hemin-impregnated disks (Sigma-Aldrich, Germany). Since variations in the thymidine content in the testing medium may affect AST results for SXT, we adhered to the EUCAST disk diffusion reading guide (v8.0). For disk diffusion and MIC gradient strip, a hazy zone of inhibition or weak growth within a clear zone of inhibition was ignored. The reading for MHF was performed according to the reading guide for standard MH agar. The reading of broth microdilution adhered to EUCAST broth microdilution reading guide v3.0, by reading the MIC at the lowest concentration that inhibits ≥80% of growth compared to the growth control. Quality control for AST methods (disk diffusion, broth microdilution, and MIC gradient strip) was performed regularly using *S. aureus* ATCC 29213 and ATCC 25923 in the routine microbiological diagnostics.

**Modified (photometric) population analysis profile.** Population analysis profile (PAP) was performed using a photometric approach. Briefly, broth microdilution was performed according to current diagnostic standards using Mueller-Hinton broth (Difco, USA) with SXT (1:19) (Sigma-Aldrich, Germany) in the concentration range of 0.06 to 32 mg/liter (referring to the TMP concentration) in a flat-bottom 96-well plate, using a similar setting as the standard broth microdilution (31). Stock SXT solution was dissolved in dimethyl sulfoxide (DMSO), and there was no antibacterial effect of DMSO (highest concentration, 1% for 32 mg/liter SXT). After 18 h of incubation, bacterial growth was semiquantified by determining the optical density at 590 nm (OD$_{590}$). To check for thymidine dependency, the mPAP was also performed in Mueller-Hinton broth, which was supplemented with 5 IU/ml thymidine phosphorylase (Sigma-Aldrich, Germany) and incubated for 24 h at 37°C to deplete the thymidine. The thymidine phosphorylase concentration of 5 IU/ml was chosen based on an estimation of the thymidine phosphorylase activity in bacterial growth medium supplemented with 5% lysed horse blood, as previously described by Ferone et al. (14). An SXT-resistant *S. aureus* isolate (D2201) and an SXT-susceptible *S. aureus* isolate (ATCC 25923) were used as controls.

**Auxotrophy testing.** Phenotypic testing for thymidine, menadione, and hemin auxotrophy was performed using impregnated disks on Mueller-Hinton agar as described by Maduka-Ezeh et al. (32). Colony growth and morphology were inspected after overnight incubation at 37°C.

***Galleria mellonella* infection model.** The wax moth larva infection model of Desbois and Coote was adapted to test for the efficacy of SXT in preventing larval mortality after infection with viable *S. aureus* (29). Batches of wax moth larvae (*Galleria mellonella*) (TZ Terraristik, Germany) were stored in the dark at room temperature (±20°C) for up to 3 days upon receipt. Since the larval size was nonuniform, only similarly sized larvae (typical mass, ~250 mg) were used for the experiments. TMP and SMZ were purchased from Sigma-Aldrich (Germany) and dissolved at a 1:19 ratio, which is the typical ratio of TMP to SMZ measured in the body (33), in sterile DMSO. Antibiotic was administered at the usual dosage for human therapy of 10 mg/kg body mass. Viable *S. aureus* bacteria in the mid-log phase grown in tryptic soy broth at 37°C with gentle shaking were washed once with phosphate-buffered saline (PBS) and adjusted to the desired concentration using the McFarland turbidity standard. Experiments were performed 3 times on different occasions with 10 larvae in each group. Each experiment batch was performed with three negative controls: (i) nonmanipulated control, (ii) PBS (trauma) control, and (iii) PBS-SXT (trauma and DMSO) control. The infection was performed using SXT-resistant isolate *S. aureus* D2201, two SXT-heteroresistant isolates, KE7160 and D2198, and ATCC 25923 as an SXT-susceptible control. Each larva was infected using an inoculum of 5 × 10$^6$ CFU, injected into the last proleg. Two hours after initial infection, all larvae were confirmed alive and then treated with either PBS or SXT. Larvae were stored in plastic petri dishes at 37°C in the dark. Larvae were inspected every 12 h for up to 36 h. Antibiotic was administered as a single dose (2 h p.i.). Dead larvae were defined as touch nonresponsive and melanized (black).

**Statistical analysis.** Descriptive statistics were performed using Stata 13 software (StataCorp, USA). Statistical testing and visualization of survival analysis (Kaplan-Meier) were performed using PRISM v9 (GraphPad, USA). For survival analysis figures, data from the three experiments were pooled to achieve $n = 30$ for each group, and statistical comparison between groups was performed using the log rank test.

## SUPPLEMENTAL MATERIAL

Supplemental material is available online only.
**SUPPLEMENTAL FILE 1**, PDF file, 0.6 MB.

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
