## [Reviewer comments · Microbiology Spectrum]

**Microbiology
Spectrum**

Phenotypic detection of hemin-inducible trimethoprim-sulfamethoxazole heteroresistance in *Staphylococcus aureus*

Dennis Nurjadi, Quan Chanthalangsy, Elfi Zizmann, Vanessa Stuermer, Maximilian Moll, Sabrina Klein, Sébastien Boutin, Klaus Heeg, and Philipp Zanger

Corresponding Author(s): Dennis Nurjadi, Heidelberg University Hospital

Review Timeline:

Submission Date:

September 10, 2021

Accepted:

September 17, 2021

Editor: Kunyan Zhang

Reviewer(s): The reviewers have opted to remain anonymous.

Transaction Report:

DOI: <https://doi.org/10.1128/Spectrum.01510-21>

September 17, 2021

Dr. Dennis Nurjadi
Heidelberg University Hospital
Department of Infectious Diseases (Medical Microbiology and Hygiene)
Im Neuenheimer Feld 324
Heidelberg
Germany

Re: Spectrum01510-21 (Phenotypic detection of hemin-inducible trimethoprim-sulfamethoxazole heteroresistance in *Staphylococcus aureus*)

Dear Dr. Dennis Nurjadi:

I have reviewed the previous reviewers' comments from JCM, your responses and your revised manuscript. I think you have addressed all the comments and concerns. Your work has merit. I decide to accept your manuscript in the current format to be published in Microbiology Spectrum.

I am forwarding it to the ASM Journals Department for publication. You will be notified when your proofs are ready to be viewed.

Sincerely,

Kunyan Zhang
Editor, Microbiology Spectrum
